# MA-RAG: Multi-Agent Retrieval-Augmented Generation via Collaborative Chain-of-Thought Reasoning

## Abstract

We present MA-RAG, a Multi-Agent framework for Retrieval-Augmented Generation (RAG) that addresses the inherent ambiguities and reasoning challenges in complex information-seeking tasks. Unlike conventional RAG methods that rely on end-to-end fine-tuning or isolated component enhancements, MA-RAG orchestrates a collaborative set of specialized AI agents: Planner, Step Definer, Extractor, and QA Agents, each responsible for a distinct stage of the RAG pipeline. By decomposing tasks into subtasks such as query disambiguation, evidence extraction, and answer synthesis, and enabling agents to communicate intermediate reasoning via chain-of-thought prompting, MA-RAG progressively refines retrieval and synthesis while maintaining modular interpretability. Extensive experiments on multi-hop and ambiguous QA benchmarks, including NQ, HotpotQA, 2WikimQA, and TriviaQA, demonstrate that MA-RAG significantly outperforms standalone LLMs and existing RAG methods across all model scales. Notably, even a small LLaMA3-8B model equipped with MA-RAG surpasses larger standalone LLMs, while larger variants (LLaMA3-70B and GPT-4o-mini) set new state-of-the-art results on challenging multi-hop datasets. Ablation studies reveal that both the planner and extractor agents are critical for multi-hop reasoning, and that high-capacity models are especially important for the QA agent to synthesize answers effectively. Beyond general-domain QA, MA-RAG generalizes to specialized domains such as medical QA, achieving competitive performance against domain-specific models without any domain-specific fine-tuning. Our results highlight the effectiveness of collaborative, modular reasoning in retrieval-augmented systems: MA-RAG not only improves answer accuracy and robustness but also provides interpretable intermediate reasoning steps, establishing a new paradigm for efficient and reliable multi-agent RAG.

## 1 Introduction

Recent advances in natural language processing have driven the development of Retrieval-Augmented Generation (RAG) models, which aim to enhance the factual accuracy and contextual relevance of generated text by integrating external knowledge sources (Lewis et al., 2020; Guu et al., 2020; Izacard & Grave, 2021; Lin et al., 2024). These systems address core limitations of Large Language Models (LLMs), such as outdated knowledge (Zhang et al., 2023b; Kasai et al., 2023) and poor generalization to domain-specific queries (Siriwardhana et al., 2023; Xiong et al., 2024), by retrieving top-$k$ documents from an external corpus (Formal et al., 2022; Izacard et al., 2022; Wang et al., 2022a) to ground the model's output in relevant evidence.

Prior research in RAG has largely concentrated on optimizing three key components—retrieval, augmentation, and generation (Gao et al., 2024; Fan et al., 2024) (Figure 1(a)). Retrieval strategies span sparse methods (Jones, 1972; Robertson & Zaragoza, 2009) and dense retrieval (Reimers & Gurevych, 2019; Karpukhin et al., 2020), each with respective weaknesses such as lexical gaps (Berger et al., 2000) or retrieval failure on out-of-distribution and multi-hop queries (Dai et al., 2023). Augmentation methods often rely on post-retrieval processing such as re-ranking or document summarization (Chen et al., 2020; Glass et al., 2022; Ma et al., 2024) (Figure 1(b)) to improve input quality for the LLM, but add latency and may still fail to filter irrelevant or misleading evi-

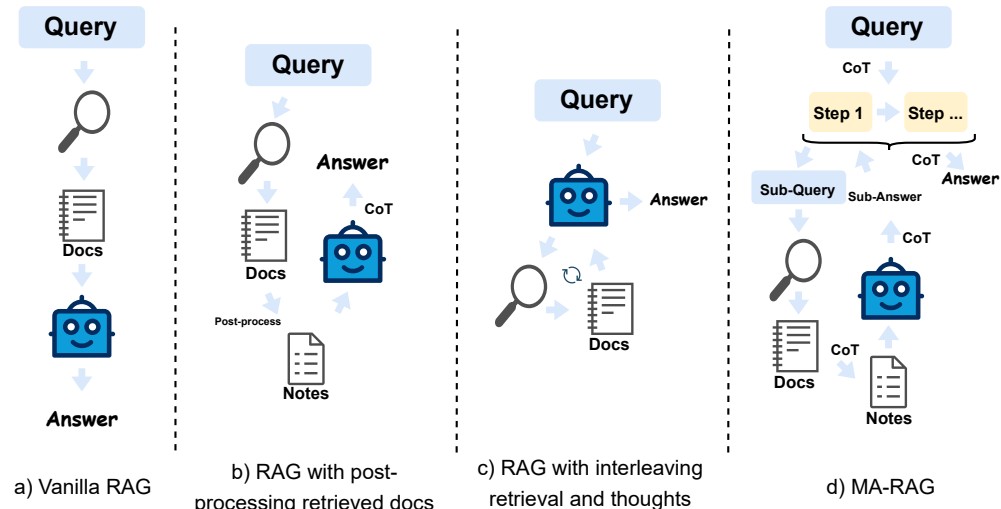

a) Vanilla RAG

b) RAG with post-processing retrieved docs

c) RAG with interleaving retrieval and thoughts

d) MA-RAG

Figure 1: **Architectural Comparison of MA-RAG and Prior RAG Methods. a)** A naive RAG system performs one-shot retrieval followed by direct answer generation. **b)** Enhanced systems incorporate post-retrieval processing such as document re-ranking or summarization. **c)** Iterative systems interleave retrieval and reasoning via query rewriting or multi-step refinement, yet often lack explicit modularity and planning. **d)** In contrast, MA-RAG adopts a collaborative multi-agent architecture where specialized agents handle distinct stages of the RAG pipeline, such as query disambiguation, targeted evidence extraction, and answer synthesis, using chain-of-thought reasoning. Agents are invoked dynamically and on demand, enabling fine-grained document analysis and step-by-step resolution of ambiguities, resulting in a more robust, interpretable, and efficient retrieval-to-generation process.

dence. More sophisticated approaches introduce iterative retrieval or query rewriting (Jiang et al., 2023b; Asai et al., 2024) (Figure 1(c)), but commonly assume the input query is well-formed and overlook the broader reasoning process across the pipeline. Across retrieval, augmentation, and generation, most existing methods treat components in isolation, failing to resolve ambiguities and reasoning gaps that span multiple stages, such as vague queries, incomplete retrievals, or scattered evidence, ultimately limiting robustness and transparency in complex QA scenarios.

To address these challenges, we propose **MA-RAG**, a modular, *training-free* Multi-Agent framework that performs step-by-step reasoning across the entire RAG pipeline (Figure 1(d)). MA-RAG views the RAG process as a collaborative effort among specialized agents, each responsible for a specific subtask such as query disambiguation and task decomposition (Planner), document retrieval (Step Definer and Retrieval Tool), evidence extraction (Extractor), and answer synthesis (QA Agent). Rather than invoking all agents uniformly, MA-RAG adopts an *on-demand* strategy, calling only the necessary agents depending on the ambiguity and complexity at each step. Each agent is guided by chain-of-thought prompting, enabling explicit intermediate reasoning that improves interpretability and task alignment. For instance, agents can decompose an underspecified query into concrete sub-questions, retrieve documents tailored to each subtask, distill targeted evidence from multiple sources, and compose coherent answers by integrating dispersed information.

This multi-agent design not only improves robustness to ambiguous and multi-hop questions, but also provides fine-grained control over the information flow in RAG, all without requiring model fine-tuning. Empirically, MA-RAG achieves new state-of-the-art performance on multiple open-domain QA benchmarks, including NQ, HotpotQA, TriviaQA, and 2WikimQA, consistently outperforming both standalone LLMs and strong RAG baselines across model scales. Ablation studies further reveal the importance of the modular architecture: the planner agent is essential for multi-hop reasoning, while the extractor agent significantly improves grounding by filtering irrelevant content. Moreover, our analysis shows that model size matters most for answer generation, planning and evidence extraction, while lighter-weight models can be effectively used for retrieval components, enabling more efficient deployments. These results highlight the effectiveness and flexibility of our multi-agent framework in tackling complex QA tasks without additional supervision or domain-specific training.

Our key contributions are as follows:

- We introduce **MA-RAG**, a modular multi-agent framework that performs reasoning-driven RAG via *structured collaboration* between agents, enabling fine-grained handling of ambiguity and complex queries.

- MA-RAG operates entirely without model fine-tuning, offering a general and adaptable solution that outperforms or matches strong baselines across multiple QA datasets and LLM backends.

- Through agent-specific chain-of-thought reasoning, MA-RAG provides *interpretable interme-diate steps* and demonstrates strong *generalization* to specialized domains, such as biomedical QA, without requiring domain-specific fine-tuning.

## 2 RELATED WORKS

**Large Language Models (LLMs)** have driven significant advancements in recent years. Starting with GPT-1 (Radford et al., 2018) on the Transformer architecture (Vaswani et al., 2017), subsequent models such as GPT-2 (Radford et al., 2019), GPT-3 (Brown et al., 2020), and GPT-4 (OpenAI, 2024) have greatly enhanced capabilities in text understanding and generation. Beyond GPT, models including Mistral (Jiang et al., 2023a), Gemini (Gemini Team, 2023), and LLaMA ((Touvron et al., 2023a), Touvron et al. (2023b)) show strong performance across tasks such as question answering and entity recognition (Zhao et al., 2023). LLM training involves unsupervised pre-training, supervised fine-tuning, and alignment with human feedback, yet domain-specific challenges remain (Kandpal et al., 2023). Techniques such as PEFT (Houlsby et al., 2019a) improve fine-tuning efficiency, while prompt-based learning (Lester et al., 2021; Li & Liang, 2021), adapters (Houlsby et al., 2019b; Fu et al., 2021; Wang et al., 2022b; He et al., 2022), and reparameterization methods (Hu et al., 2022; Edalati et al., 2022; Dettmers et al., 2023) selectively adjust parameters for improved performance. Additionally, recent studies (Wei et al., 2024; Java et al., 2025) highlight the importance of evaluating factuality and retrieval efficiency in short-form and deep research scenarios.

**Retrieval-Augmented Generation (RAG)** enhances LLM performance by integrating external knowledge via document retrieval (Lewis et al., 2020; Guu et al., 2020). Challenges remain in determining what, when, and how to retrieve (Gao et al., 2024). Early methods incorporate retrieval into next-token prediction (Khandelwal et al., 2020; Ram et al., 2023; Liu et al., 2024b) or use end-to-end pipelines (Guu et al., 2020; Borgeaud et al., 2022; Izacard et al., 2023; Zhang et al., 2024a), while others study knowledge representation and retrieval robustness (Xu et al., 2024; Sarthi et al., 2024). Supervised and contrastive approaches often face scalability and domain transfer limitations (Dai et al., 2023; Zhang et al., 2023b; Shi et al., 2024). Structured retrieval methods include HippoRAG (Gutiérrez et al., 2024), which leverages knowledge graphs. Recent research focuses on tighter retrieval-reasoning integration. RA-DIT (Lin et al., 2024) tunes LLMs for context use and retrievers for relevance independently. Speculative RAG (Wang et al., 2025) drafts answers with a specialist LM and verifies with a generalist LM. CD-LM (Li et al., 2025b) improves inference via chunk-level retrieval, while Auto-GDA (Leemann et al., 2025) addresses domain adaptation with synthetic data. Graph-based methods such as ToG-2 (Ma et al., 2025) and SubgraphRAG (Li et al., 2025a) utilize subgraph structures to enhance retrieval. Reinforcement learning approaches optimize retrieval and generation via episodic memory (Shinn et al., 2023), policy optimization (Kulkarni et al., 2024), evidence citation (Menick et al., 2022), and reflection-based refinement (Asai et al., 2024; Zhou et al., 2023; Gao et al., 2025). Recent open-source agentic retrieval frameworks, such as Open Deep Search (Alzubi et al., 2025a), further demonstrate the potential of lightweight reasoning agents for democratized, structured search. Compared to prior agent-based RAG systems, our MA-RAG provides a training-free, efficient, and interpretable solution.

**LLM-based Agentic Systems** coordinate multiple specialized agents to solve complex tasks through structured interaction (Guo et al., 2024). Agents operate in diverse environments, including sandboxed, physical, or abstract settings (Hong et al., 2024; Mao et al., 2025; Park et al., 2023), and assume predefined, emergent, or data-driven roles (Du et al., 2024; Xiong et al., 2023). Communication follows cooperative, competitive, or debate-based paradigms through centralized or decentralized channels (Liu et al., 2024c; Hong et al., 2024). Capabilities are developed via environmental feedback, memory retrieval, or self-evolution (Wang et al., 2023; 2024; do Nascimento et al., 2023; Zhang et al., 2023a; Chen et al., 2024a;b). Recent work has extended agentic designs to RAG. For example, Agentic RAG for time series (Ravuru et al., 2024) uses hierarchical agent routing, while

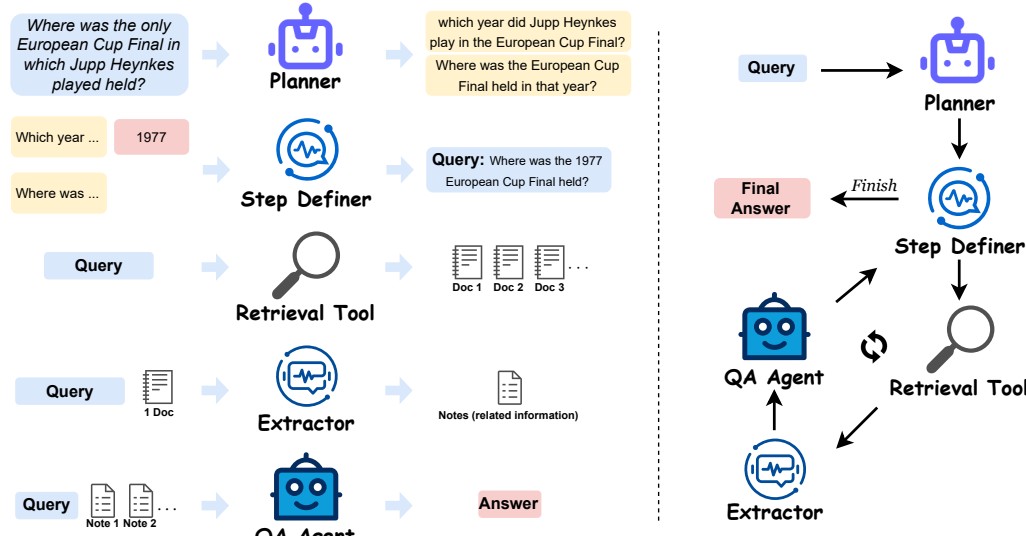

Figure 2: **Overview of MA-RAG.** MA-RAG is a training-free, multi-agent RAG framework that decomposes complex queries into interpretable steps through collaborative reasoning. The left panel shows individual components and their I/O interfaces; the right panel illustrates the overall iterative workflow. A **Planner Agent** first breaks down the input query into a high-level reasoning plan. For each step, a **Step Definer Agent** generates a detailed subquery based on the step goal, original question, and prior outputs. This subquery is processed by the **Retrieval Tool** to fetch top-ranked documents, which are then refined by the **Extractor Agent** to retain only step-relevant content. The **QA Agent** synthesizes the final answer for each step using the filtered evidence and subquery. MA-RAG iterates through these steps until the full reasoning path is complete.

CollEX (Schneider et al., 2025) enables multimodal retrieval via vision-language agents. MA-RAG differs by employing lightweight, specialized agents that collaborate through chain-of-thought reasoning, improving transparency and performance in complex QA settings without fine-tuning.

## 3 METHOD

In this section, we introduce MA-RAG, our proposed multi-agent framework for retrieval-augmented generation. We begin by formalizing the RAG problem setting, and then describe our multi-agent approach designed to improve both retrieval and reasoning.

**Preliminaries** Retrieval-augmented generation leverages a large corpus of documents or contexts—such as Wikipedia—to provide grounded knowledge for question answering. Given a query $q$ and a corpus $\mathcal{C}$, a dense retriever $\mathcal{R}$ retrieves the top-$k$ relevant contexts $C_q = \{c_1, ..., c_k\}$. In standard RAG pipelines, a large language model (LLM) generates the answer based on a prompt that includes the query and the retrieved documents:

$$y = \text{LLM}(\text{Prompt}_{\text{gen}}(q, C_q)),$$

where $\text{Prompt}_{\text{gen}}$ is a prompting template that provides instructions and structures the input for the LLM. This paradigm allows the LLM to produce answers grounded in retrieved evidence, thereby reducing hallucinations (Huang et al., 2023).

### 3.1 MULTI-AGENT SYSTEM FOR RETRIEVAL-AUGMENTED GENERATION (MA-RAG)

While advances in long-context LLMs (Liu et al., 2025) suggest potential to bypass retrieval altogether, practical limitations remain: effective context utilization is still far below advertised limits (Modarressi et al., 2025), and processing long sequences significantly increases inference cost and latency. More importantly, RAG is not merely a workaround for context size—it is a framework for extending LLMs' factual coverage by dynamically incorporating external knowledge. We emphasize a system-level perspective: RAG should be treated as a pipeline for complex, knowledge-intensive reasoning, not just improved generation.

Two persistent challenges degrade RAG performance. First, *retrieval mismatch* arises from semantic gaps between user queries and corpus content due to ambiguity, domain shift, or granularity differences. Second, *context inefficiency* stems from naïvely appending all retrieved passages, which inflates input length and model attention without guaranteeing relevance (Liu et al., 2024a). Moreover, document chunking introduces trade-offs: larger chunks preserve context but increase noise; smaller ones lose coherence.

To address these challenges, we propose **MA-RAG**, a lightweight, training-free multi-agent RAG framework that decomposes complex queries into structured reasoning steps and coordinates specialized agents for high-precision retrieval and generation. Figure 2 presents an overview of the system, which includes four collaborating agents and one retrieval module.

**Planner Agent.** The Planner analyzes the input query $q$ to perform query disambiguation and task decomposition. It identifies ambiguities or underspecified elements and reformulates them into clearer sub-questions if needed. For complex or multi-hop queries, it produces a structured plan $P = \{s_1, s_2, \ldots, s_n\}$, where each $s_i$ denotes a reasoning subtask. The number of reasoning steps is determined by the Planner, and at each step the system performs a Retrieve $\rightarrow$ Answer RAG process using a sub-query refined by the Step Definer. Breaking down questions into simpler, targeted sub-queries enables more precise retrieval, and our empirical results across benchmarks validate this design. The Planner is guided by chain-of-thought prompting with few-shot examples, ensuring interpretable step-wise decomposition that supports grounded reasoning in downstream modules.

**Step Definer Agent.** Each abstract step $s_i$ is made executable by the step definer, which generates a detailed subquery tailored for retrieval. This agent conditions on the original query $q$, the plan $P$, the current step $s_i$, and accumulated history $H_{i-1} = \{(s_1, a_1), \ldots, (s_{i-1}, a_{i-1})\}$. By grounding the subquery in context and prior answers, the step definer bridges high-level intent and low-level execution, enabling precise and relevant document retrieval.

**Retrieval Tool.** We use a dense retrieval module built on FAISS (Johnson et al., 2021) for fast, scalable search over large corpora. Texts are preprocessed into chunks and embedded using a pretrained encoder. At inference, the subquery is encoded into a vector and matched against the index via inner product. The top-$k$ relevant passages are returned, enabling dynamic, on-demand knowledge augmentation at each step.

**Extractor Agent.** Retrieved passages often contain redundant or irrelevant content. Instead of appending entire chunks, the Extractor selects and aggregates sentences or spans directly aligned with the current subquery. This not only filters out noise and mitigates the lost-in-the-middle issue (Liu et al., 2024a), but also enables effective evidence aggregation by combining complementary information from multiple sources into a concise evidence set for the QA agent. To avoid context overflow in multi-hop queries, the Extractor summarizes relevant content at each step, and only the step-level query with the extracted summary or answer is passed forward. This preserves continuity while keeping context concise and efficient, ultimately supporting more accurate and informed answer generation.

**Question Answering Agent.** Given the step-specific query and filtered evidence, the QA agent synthesizes an answer using in-context learning. It produces a response $a_i$ for each step $s_i$, which is passed to the next iteration. Once all steps are completed, the final answer is assembled and returned to the user.

A key feature of MA-RAG is its dynamic and modular agent invocation. Rather than executing a fixed pipeline, the system orchestrates agents on demand based on the structure of the reasoning plan. The **Planner Agent** is invoked once at the beginning to generate a high-level plan. Subsequently, for each step $s_i$, the system triggers the **Step Definer Agent** to produce a detailed subquery, which is then passed to the **Retrieval Tool** and the **Extractor Agent** in sequence. The extracted evidence is sent to the **QA Agent**, which returns the answer $a_i$. This answer is added to the history $H_i$, and the next iteration begins. The system maintains state throughout the reasoning trajectory, allowing each agent to condition on the evolving context. This modular design enables flexible, step-by-step execution and supports adaptive reasoning over complex, multi-hop queries without requiring all agents to be active simultaneously. The complete implementation and agent communication details are provided in Appendix.

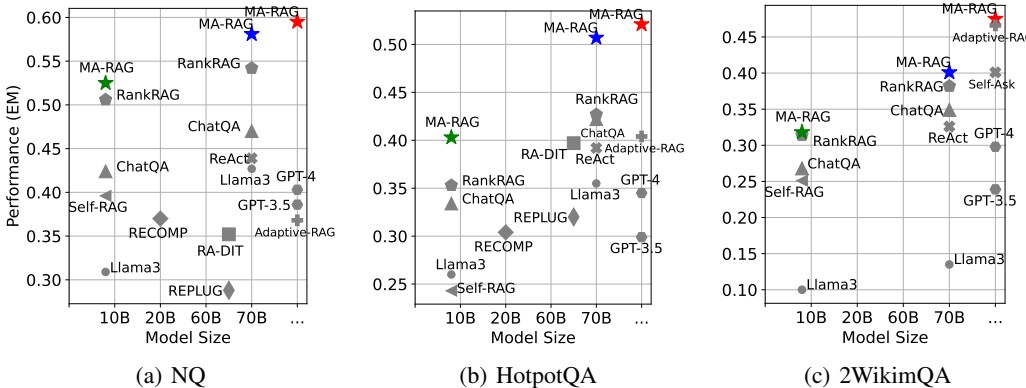

(a) NQ  (b) HotpotQA  (c) 2WikimQA

Figure 3: Exact Match (EM) performance of MA-RAG and baseline methods on NQ, HotpotQA, and 2WikimQA. The green star indicates MA-RAG with LLaMA3-8B, the blue star indicates MA-RAG with LLaMA3-70B, and the red star indicates MA-RAG with GPT-4o-mini. Across all datasets, MA-RAG consistently outperforms baseline methods using the same model size, demonstrating the effectiveness of our multi-agent reasoning approach.

## 4 EXPERIMENTS

**Datasets.** We evaluate MA-RAG on two tasks: *Open-domain Question Answering* and *Fact Verification*. For open-domain QA, we use four widely adopted benchmarks: Natural Questions (NQ) (Kwiatkowski et al., 2019), TriviaQA (Joshi et al., 2017), HotpotQA (Yang et al., 2018), and 2WikimQA (Ho et al., 2020). Among them, NQ and TriviaQA consist primarily of single-hop questions, while HotpotQA and 2WikimQA require multi-hop reasoning across multiple evidence sources. For fact verification, we use FEVER (Thorne et al., 2018) as our primary benchmark.

While we include results on TriviaQA and FEVER for completeness, these datasets may be suboptimal for evaluating the effectiveness of RAG methods. In particular, GPT-4 can already achieve strong performance on these benchmarks, 84.8 EM and 87.7 Acc respectively, even without retrieval augmentation. This is because their questions often do not require external knowledge retrieval, making them less suitable for assessing the benefits of retrieval-augmented approaches.

**Baselines.** For question-answering, we consider baseline LLMs without RAG include GPT-3.5-turbo (OpenAI, 2022), GPT-4 (OpenAI, 2024), Llama3-Instruct 8B (Meta, 2024), and Llama3-Instruct 70B (Meta, 2024). We also consider baselines in RAG include Atlas (Izacard et al., 2023), Recomp (Xu et al., 2024), Replug (Shi et al., 2024), Ra-dit (Lin et al., 2024), Self-RAG (Asai et al., 2024), ChatQA-1.5 (Liu et al., 2024b), RankRAG (Yu et al., 2024)[1], Adaptive-RAG (Jeong et al., 2024), ReAct (Yao et al., 2023), Self-Ask (Press et al., 2023), and Smart-RAG (Gao et al., 2025).

**Evaluation Metrics.** For *Open-domain QA* tasks, we use *Exact Match (EM)* as the main metric for comparison, while we use *Accuracy (Acc)* for *Fact verification*.

**Implementation Details.** For NQ, HotpotQA, TriviaQA, and FEVER, we use the split from KILT benchmark (Petroni et al., 2021). We use Wikipedia corpus preprocessed by Karpukhin et al. (2020). We use gte-multilingual (Zhang et al., 2024b) as our retrieval model. We use different LLMs to build MA-RAG in different size, from small LLMs with 8B parameters (Meta, 2024) to middle size LLMs with 70B parameters (Meta, 2024) to black box LLMs (OpenAI, 2024).

### 4.1 RESULTS

Figure 3 provides a visual comparison between MA-RAG and several baseline models across multiple datasets. The full numerical results with additional discussion are available in the Appendix. Key observations from our experiments include:

**MA-RAG outperforms standalone LLMs without retrieval.** Our results show that MA-RAG significantly enhances the performance of base LLMs when combined with retrieval-augmented

---

[1]By the time of submission, RankRAG had not released any models. The reported results are therefore directly copied from their paper.

Table 1: Ablation study with MA-RAG using 70B LLM: evaluating the impact of planner and extractor agents on MA-RAG performance across single-hop and multi-hop QA benchmarks.

| Task | NQ | TriviaQA | HotpotQA | 2WikimQA | FEVER |
|------|------|----------|----------|----------|-------|
| MA-RAG (Llama3-70B) | 58.1 | 85.4 | 50.7 | 43.1 | 93.1 |
| - w/o Extractor | 53.4 | 82.1 | 43.4 | 38.2 | 89.2 |
| - w/o Planer | 57.9 | 80.3 | 36.2 | 26.4 | 91.3 |

reasoning. For instance, while Llama3-70B and GPT-4 achieve accuracy scores of 42.7 and 40.3 on NQ, respectively, MA-RAG (Llama3-8B) already surpasses these models with a score of 52.5, and MA-RAG (GPT-4o-mini) achieves an even higher score of 59.5. Similar improvements are observed across other datasets, including HotpotQA and 2WikimQA, where MA-RAG demonstrates a substantial advantage in handling complex, knowledge-intensive questions. These results underline that retrieval-augmented reasoning, supported by a multi-agent framework, outperforms standalone LLMs that rely solely on their internal knowledge base.

**MA-RAG outperforms existing RAG models.** At the 8B scale, MA-RAG consistently outperforms several strong baseline models, such as ChatQA-1.5 8B and RankRAG 8B. Despite using models of comparable size, MA-RAG (Llama3-8B) achieves superior exact match (EM) scores on NQ, HotpotQA, and 2WikimQA, showcasing the effectiveness of our multi-agent architecture in optimizing retrieval and reasoning. Even when compared to larger retrieval models like RA-DIT 65B and REPLUG 65B, MA-RAG (8B) demonstrates consistently better performance across all tasks, indicating that our approach is more effective at leveraging external knowledge while maintaining efficiency.

When scaling up to larger models, MA-RAG (Llama3-70B and GPT-4o-mini) outperforms the strongest 70B-scale models, such as ChatQA-1.5 70B and RankRAG 70B, setting new state-of-the-art results on multiple benchmarks. Notably, MA-RAG achieves a score of 59.5 on NQ, 87.2 on TriviaQA, 52.1 on HotpotQA, and 47.5 on 2WikimQA. In particular, on more challenging, multi-hop, and long-tailed datasets like HotpotQA and 2WikimQA, MA-RAG demonstrates significant gains over previous methods. These improvements suggest that the fine-grained query decomposition and passage extraction capabilities inherent in MA-RAG are particularly advantageous in handling complex retrieval conditions. A key strength of our modular design is that the number of steps in MA-RAG is dynamically determined by the planner based on question complexity, with multi-hop questions resulting in more steps and LLM calls. For example, on HotpotQA, MA-RAG averages 2.3 steps per question, while on NQ, which is mostly single-hop, it averages 1.4 steps. Overall, these results highlight the critical role of multi-agent coordination in improving open-domain QA performance, emphasizing that the integration of specialized agents for different reasoning steps leads to more effective and efficient utilization of external knowledge sources.

## 4.2 ABLATION STUDY

**Impact of Agents.** To understand each agent's contribution, we conduct an ablation study by removing either the Extractor or Planner from MA-RAG. Table 1 reports performance across five QA benchmarks. Without the Extractor, retrieved documents are fed directly into the prompt, leading to consistent performance drops and highlighting its role in refining input and grounding responses. Removing the Planner reduces MA-RAG to a single-turn RAG system with document filtering but no query decomposition. While it still performs well on simpler, single-hop datasets, it struggles with multi-hop questions requiring structured reasoning. The largest performance drop occurs on multi-hop datasets, emphasizing the Planner's importance in guiding complex reasoning. These results show both agents are essential: the Extractor enhances precision, and the Planner enables effective reasoning across diverse question types.

**Impact of LLMs.** To assess the effect of model size on different agents in our multi-agent system, we conduct an ablation study where each agent is individually replaced with an LLaMA3-8B model while keeping the others as LLaMA3-70B. This isolates the impact of LLM capacity across agents on two multi-hop datasets: HotpotQA and 2WikimQA (Table 2). Replacing the QA agent consistently causes the largest performance drop, especially on 2WikimQA, suggesting that high-capacity models are crucial for final answer generation. Substituting the Planner or Extractor also leads to clear declines, with the Extractor highlighting the challenge of identifying relevant evidence

Table 2: Ablation study on LLMs' size: evaluating the impact of replacing individual agents with LLama3-8B in a 70B-based MA-RAG system on multi-hop QA.

| Planner | Step definer | Extractor | QA | HotpotQA | 2WikimQA |
|---|---|---|---|---|---|
| Llama3-70B | Llama3-70B | Llama3-70B | Llama3-70B | 50.7 | 43.1 |
| Llama3-70B | Llama3-70B | Llama3-70B | Llama3-8B | 49.7 | 34.5 |
| Llama3-70B | Llama3-70B | Llama3-8B | Llama3-70B | 49.4 | 39.8 |
| Llama3-70B | Llama3-8B | Llama3-70B | Llama3-70B | 49.9 | 42.5 |
| Llama3-8B | Llama3-70B | Llama3-70B | Llama3-70B | 49.2 | 39.1 |

in complex retrieval. The Planner's ability to generate effective reasoning plans is similarly sensitive to model capacity. In contrast, reducing the Step Definer has only marginal impact, indicating its structured role is less dependent on large models.

In summary, for multi-hop QA tasks like HotpotQA and 2WikimQA, it is critical to allocate larger models to the QA, planner and extractor agents to maintain performance. Smaller models can be used for step definer with minimal loss, enabling more efficient resource allocation in practice.

**Other Domains** We further assess the generalizability of our method by conducting experiments in other domains. Specifically, we evaluate MA-RAG on medical benchmark datasets, including PubmedQA and MedMCQA. We follow the setup in Xiong et al. (2024), using Med-CPT (Jin et al., 2023) as the retrieval model and deploying MedCorp (Xiong et al., 2024) as the corpus.

The experimental results of MA-RAG and the baselines are shown in Table 3. We compare MA-RAG with various RAG baseline models, including Mixtral (Jiang et al., 2024), Llama2-70B (Touvron et al., 2023b), Meditron-70B (Chen et al., 2023), PMC-Llama 13B (Wu et al., 2024), ChatQA-1.5 (Liu et al., 2024b), RankRAG (Yu et al., 2024), GPT-3.5 (OpenAI, 2022), and GPT-4-0613 (OpenAI, 2024), under

Table 3: The accuracy of MA-RAG and baselines on medical benchmark datasets. All of baselines use retrieval in the same settings. We collect results for baselines from public reports (Xiong et al., 2024).

| Method | PubmedQA | MedMCQA |
|---|---|---|
| Mixtral 8*7B (2024) | 67.6 | 56.4 |
| Llama2 70B (2023b) | 50.4 | 43.1 |
| Meditron 70B (2023) | 56.4 | 52.7 |
| PMC-llama 13B (2024) | 42.6 | 65.2 |
| ChatQA-1.5 8B (2024b) | 66.4 | 46.9 |
| ChatQA-1.5 70B (2024b) | 74.8 | 62.5 |
| RankRAG 8B (2024) | 65.0 | 56.9 |
| RankRAG 70B (2024) | 79.8 | 69.1 |
| GPT-3.5 (2022) | 67.4 | 66.7 |
| GPT-4-0613 (2024) | 70.6 | 66.7 |
| MA-RAG (Llama3-8B) | 66.7 | 56.5 |
| MA-RAG (Llama3-70B) | 78.9 | 67.9 |
| MA-RAG (GPT-4o-mini) | 80.2 | 69.8 |

identical settings. MA-RAG demonstrates strong performance in the medical domain, despite *not being fine-tuned on biomedical data*. Notably, MA-RAG with Llama3-70B outperforms domain-specific models such as Meditron 70B and PMC-LLaMA 13B, achieving performance comparable to GPT-4. When using GPT-4o-mini, MA-RAG surpasses all baselines, including GPT-4-0613 and RankRAG 70B. These results underscore the generalizability of MA-RAG to specialized domains through modular reasoning and chain-of-thought coordination, *without the need for domain-specific fine-tuning*.

**Case Study** Table 4 presents a case study from the 2WikimQA dataset. Given a complex query, our model first generates a plan and solves the problem step by step. Even though each sub-query is detailed and single-hop, the retrieved documents remain noisy, and the extractor agent selectively retains only the relevant information. To ensure a fair comparison, we report results generated exclusively by 70B models. Our main results are based on Llama3-70B, and for reference, we also include direct answers from Llama3-70B without RAG. Comparisons with RankRAG are not provided, as it is not open source. Additional examples can be found in the Appendix.

**MA-RAG with Internet Access.** We further evaluated MA-RAG with web access by integrating Google Search as the retrieval engine on the SimpleQA benchmark (Wei et al., 2024). Results in the Appendix show that MA-RAG achieves strong performance (GPT-4o: 40.1% versus MA-RAG (GPT-4o-mini, web): 86.4%), demonstrating its flexibility and competitive accuracy among recent web-enabled systems.

**Discussions.** MA-RAG's multi-agent design, while improving reasoning and interpretability, introduces additional runtime and token overhead. Each agent invocation involves separate prompts

Table 4: A case study on the 2WikiMQA dataset. The blue texts represent the ground truth answer or correct outputs, while red texts denote distractors or wrong outputs, and green texts stand for evidences. Note that none of the retrieved documents contain the correct answer directly. All experiments are conducted under the same setting, using the same corpus (Karpukhin et al., 2020) and the same retrieval model (Zhang et al., 2024b).

| | |
|---|---|
| **Q**: Who is Edward De Vere, 17Th Earl Of Oxford's paternal grandfather? **Correct Answer**: John de Vere, 15th Earl of Oxford | |
| **MA-RAG 70B** | **Plan**: ["Identify Edward De Vere, 17th Earl of Oxford's father", "Determine the name of Edward De Vere, 17th of Oxford's paternal grandfather based on the father's name."] 
 **Step 1** (Identify Edward De Vere, 17th Earl of Oxford's father) 
 **Step query**: Who is Edward De Vere, 17th Earl of Oxford's father? 
 **Retrieved DocIDs**: 129772, 6127861, 129773, 14010350, 12540430 
 **Extracted Information**: *doc-129772*: [No related information about his father.], *doc-6127861*: [No related information about his father.], *doc-129773*: [Edward De Vere, 17th Earl of Oxford's father is John de Vere, 16th Earl of Oxford. This is evidenced by the passage stating, "De Vere was the only son of John de Vere, 16th Earl of Oxford."], *doc-14010350*: [No related information about his father.], *doc-12540430*: [No related information about his father.] 
 **Step Answer**: The question asks for the identity of Edward De Vere's father. The relevant document doc-129773 clearly states that Edward De Vere, 17th Earl of Oxford, is the son of John de Vere, 16th Earl of Oxford. Other documents do not provide additional information about his father. 
 **Step 2** (Determine the name of Edward De Vere, 17th Earl of Oxford's paternal grandfather based on the father's name.) 
 **Step query**: Who was the father of John de Vere, 16th Earl of Oxford? 
 **Retrieved DocIDs**: 6127862, 6127858, 17870603, 12540430, 14010350 
 **Extracted Information**: *doc-6127862*: [John de Vere, 16th Earl of Oxford was born to John de Vere, 15th Earl of Oxford.], *doc-6127858*: [John de Vere, 16th Earl of Oxford (1516 – 3 August 1562) was born to John de Vere, 15th Earl of Oxford.], *doc-17870603*: [No related information], *doc-14010350*: [No related information about his father.], *doc-14010350*: [No related information about his father.] 
 **Step Answer**: The relevant documents indicate that John de Vere, 15th Earl of Oxford is the father of John de Vere, 16th Earl of Oxford 
 **Answer**: John de Vere, 15th Earl of Oxford |
| **Llama3-70B** | **Answer**: Edward De Vere, 17th Earl of Oxford's paternal grandfather is John De Vere, 13th Earl of Oxford. |
| **ChatQA 70B** | **Query**: Who is Edward De Vere, 17Th Earl Of Oxford's paternal grandfather? 
 **Retrieved Docs**: 
 **Doc-129772**: Edward de Vere, 17th Earl of Oxford Edward de Vere, 17th Earl of Oxford (; 12 April 155024 June 1604) was an English peer and courtier of the Elizabethan era. Oxford was heir to the second oldest earldom in the kingdom, a court favourite for a time, ... 
 **Doc-17870603**: De Vere family The de Vere family were an English aristocratic family who derived their surname from Ver (department Manche, commune Coutances, canton Gavray), in Lower Normandy, France. The family's Norman founder in England, Aubrey (Albericus) de Vere ... 
 **Doc-6127861**: son, Edward de Vere, 17th Earl of Oxford, and a daughter, Mary de Vere. Margery died on 2 December 1568. After his death in Oxford, he was buried in Castle Hedingham, Essex, on 31 August 1562... 
 **Doc-129778**: was placed in the household of Sir William Cecil, her secretary of state and chief advisor. At 12, de Vere had become the 17th Earl of Oxford, Lord Great Chamberlain of England, and heir to an estate whose annual income ... 
 **Doc-129773**: proposed for the authorship of Shakespeare's works. De Vere was the only son of John de Vere, 16th Earl of Oxford, and Margery Golding. After the death of his father in 1562, he became a ward of Queen Elizabeth and was sent to live in the household ... 
 **Answer**: John de Vere, 16th Earl of Oxford |
| **ReAct (70B)** | **Query**: Who is Edward De Vere, 17Th Earl Of Oxford's paternal grandfather? 
 **AI Messages**: Retrieve(query=paternal grandfather of Edward De Vere, 17th Earl of Oxford) 
 **Retrieve documents**: 
 **Doc-129772**: Edward de Vere, 17th Earl of Oxford Edward de Vere, 17th Earl of Oxford (; 12 April 155024 June 1604) was an English peer and courtier of the Elizabethan era. Oxford was heir to the second oldest earldom in the kingdom, a court favourite for a time ... 
 **Doc-6127861**: son, Edward de Vere, 17th Earl of Oxford, and a daughter, Mary de Vere. Margery died on 2 December 1568. After his death in Oxford, he was buried in Castle Hedingham, Essex, on 31 August 1562. The Earl was known as a sportsman, ... 
 **Doc-17870603**: De Vere family The de Vere family were an English aristocratic family who derived their surname from Ver (department Manche, commune Coutances, canton Gavray), in Lower Normandy, France. The family's Norman founder in England, Aubrey (Albericus) de Vere... 
 **Doc-129773**: proposed for the authorship of Shakespeare's works. De Vere was the only son of John de Vere, 16th Earl of Oxford, and Margery Golding. After the death of his father in 1562, he became a ward of Queen Elizabeth and was sent to ... 
 **Doc-6393404**: Kirke and granddaughter of Aurelian Townshend. They had five children: Since he had no surviving sons and as no other suitable claimant came forward, he became the last de Vere Earl of Oxford, one of the longest-lived titles in the peerage of England; the first de Vere ... 
 **Answer**: John de Vere, 16th Earl of Oxford , was the paternal grandfather of Edward De Vere, 17th Earl of Oxford. |

and responses, which can increase latency and inference cost, especially for complex queries requiring multiple reasoning steps. Although agents are called on demand to reduce unnecessary computation, the workflow remains more resource-intensive than single-pass or standalone RAG systems. On single-hop datasets, MA-RAG achieves an average response time of about 2.2 seconds using GPT-4o-mini, while on multi-hop questions the response time increases to about 4.1 seconds. We believe this latency remains acceptable for real-world use, especially given the significant performance gains achieved. Our ablation study also shows that not all agents require large models, suggesting that assigning resource-efficient agents to different subtasks is a promising direction.

## 5 CONCLUSION

We present MA-RAG, a novel multi-agent framework for Retrieval-Augmented Generation that recasts the RAG pipeline as a coordinated reasoning process among specialized agents. Each agent, responsible for planning, retrieval, extraction, or generation, uses chain-of-thought prompting to enable structured and context-sensitive query resolution. Unlike prior work that tunes individual components or requires supervised training, MA-RAG is entirely training-free and generalizes well across domains and question types. Empirical results on five open-domain and multi-hop QA benchmarks show that MA-RAG outperforms strong LLMs and state-of-the-art RAG baselines, achieving new best results on several datasets. Ablation studies confirm the importance of the planner and extractor: the former decomposes complex queries, while the latter improves retrieval precision. Strategic allocation of model capacity across agents yields further gains in both performance and efficiency. Together, these findings highlight the potential of modular, agent-based reasoning as a scalable and adaptable approach to improving retrieval-augmented generation.

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

# A  APPENDIX

## A.1  DETAILS OF EXPERIMENTAL DATASET

### A.1.1  MAIN EXPERIMENTS

We evaluate our method on five publicly available QA benchmarks that cover both single-hop and multi-hop reasoning.

- **Natural Questions (NQ)** (Kwiatkowski et al., 2019) consists of real user queries from Google Search, where answers are short spans extracted from Wikipedia articles. We use 2837 questions from the development set in the KILT benchmark (Petroni et al., 2021) for evaluation.

- **TriviaQA** (Joshi et al., 2017) includes challenging trivia questions written by trivia enthusiasts, paired with independently collected evidence documents. We use 5359 questions from the development set in the KILT benchmark (Petroni et al., 2021) for evaluation.

- **HotpotQA** (Yang et al., 2018) is a multi-hop QA dataset that requires reasoning over multiple Wikipedia articles to answer complex questions. We use 5600 questions from the development set in the KILT benchmark (Petroni et al., 2021) for evaluation.

- **2WikiMultiHopQA (2WikimQA)** (Ho et al., 2020) is a multi-hop dataset featuring questions grounded in two distinct Wikipedia entities, designed to evaluate a model's ability to retrieve and reason across multiple sources.

- **FEVER** (Thorne et al., 2018) is a fact verification benchmark in which models must determine whether a claim is supported, refuted, or unverifiable based on evidence retrieved from Wikipedia. We use 10444 questions from the development set in the KILT benchmark (Petroni et al., 2021) for evaluation.

### A.1.2  MEDICAL BENCHMARKS

We use two dataset in medical domain to test our method.

- **MedMCQA** (Pal et al., 2022) is a multiple-choice QA dataset based on Indian medical entrance exams. We use its 4,183-question development set for evaluation.

- **PubmedQA** (Jin et al., 2019) is a biomedical QA dataset of 1,000 yes/no/maybe questions derived from PubMed abstracts.

## A.2 EXPERIMENT RESULTS

Table 5: **Results of MA-RAG and baselines on different datasets.** Results unavailable in public reports are marked as "–". We use NQ, TriviaQA, HotpotQA, and FEVER from the KILT benchmark (Petroni et al., 2021). We report accuracy for the FEVER dataset and exact match for the others.

| Task | NQ | TriviaQA | HotpotQA | 2WikimQA | FEVER |
|---|---|---|---|---|---|
| **Metric** | EM | EM | EM | EM | Acc |
| *Without Retrieval-augmented Generation* | | | | | |
| Llama3-Instruct 8B (2024) | 30.9 | 70.7 | 26.0 | 9.6 | 88.9 |
| Llama3-Instruct 70B (2024) | 42.7 | 82.4 | 35.5 | 13.5 | 91.4 |
| GPT-3.5-turbo-1106 (2022) | 38.6 | 82.9 | 29.9 | 23.9 | 82.7 |
| GPT-4-0613 (2024) | 40.3 | 84.8 | 34.5 | 29.8 | 87.7 |
| *With Retrieval-augmented Generation* | | | | | |
| SmartRAG 7B (2025) | - | - | 26.0 | - | - |
| Atlas 11B (2023) | 26.7 | 56.9 | 34.7 | - | 77.0 |
| RECOMP 20B (2024) | 37.0 | 59.0 | 30.4 | - | - |
| REPLUG 65B (2024) | 28.8 | 72.6 | 32.0 | - | 73.3 |
| RA-DIT 65B (2024) | 35.2 | 75.4 | 39.7 | - | 80.7 |
| Self-RAG 8B (2024) | 39.6 | 78.2 | 24.3 | 25.1 | - |
| ChatQA-1.5 8B (2024b) | 42.4 | 81.0 | 33.4 | 26.8 | 90.9 |
| ChatQA-1.5 70B (2024b) | 47.0 | 85.6 | 42.2 | 34.9 | 92.7 |
| RankRAG 8B (2024) | 50.6 | 82.9 | 35.3 | 31.4 | 92.0 |
| RankRAG 70B (2024) | 54.2 | 86.5 | 42.7 | 38.2 | **93.8** |
| ReAct (70B) (2023) | 43.9 | 84.5 | 39.2 | 32.6 | 92.0 |
| Adaptive-RAG (GPT-3.5) (2024) | 36.8 | - | 40.4 | 46.6 | - |
| Self-Ask (GPT-3) (2023) | - | - | - | 40.1 | - |
| *Ours* | | | | | |
| MA-RAG (Llama3-8B) | 52.5 | 82.6 | 40.3 | 31.8 | 91.4 |
| MA-RAG (Llama3-70B) | 58.1 | 85.4 | 50.7 | 43.1 | 93.1 |
| MA-RAG (GPT-4o-mini) | **59.5** | **87.2** | **52.1** | **47.5** | 93.3 |

MA-RAG achieves competitive performance on both TriviaQA and FEVER, with GPT-4o-mini reaching 87.2 EM and 93.3 accuracy, respectively, on par with or surpassing strong finetuned baselines such as RankRAG. Notably, unlike these methods, MA-RAG is fully training-free, relying solely on agent-based reasoning and chain-of-thought prompting without any gradient-based updates to the underlying LLMs. However, we caution that these benchmarks may not fully reflect the advantages of retrieval-augmented methods. Strong LLMs like GPT-4 already perform well without external retrieval (e.g., 84.8 EM on TriviaQA and 87.7 accuracy on FEVER), likely due to the fact that many questions are either single-hop or already aligned with the model's pretraining data. We include these results for completeness but emphasize that more complex, multi-hop datasets provide a better testbed for evaluating retrieval and reasoning capabilities.

## A.3 MA-RAG WITH INTERNET ACCESS

To further evaluate the capabilities of MA-RAG in practical information-seeking scenarios, we conducted additional experiments by granting MA-RAG access to real-time web search. Specifically, we integrated Google Search as the external retrieval engine and evaluated on the SimpleQA benchmark (Wei et al., 2024). This dataset is designed to assess factual question-answering abilities of frontier models *without* access to the web. It contains questions that often require up-to-date or obscure knowledge, making it a challenging benchmark for retrieval-augmented systems. We enabled MA-RAG to retrieve evidence via Google Search, replacing the retrieval tool with this search engine.

**Results.** As shown in Table 6, MA-RAG achieved an accuracy of 86.4% on SimpleQA, demonstrating a significant improvement over GPT-4o (40.1%) and the GPT-4o-mini baseline when equipped with web access. While DeepSeek-R1 (82.4%) performs significantly better than GPT-4o alone, our MA-RAG (GPT-4o-mini, web) results show that multi-agent reasoning substantially boosts performance. Although our current results are still below ODS-v2/ODS-v1+DeepSeek-R1* (Alzubi et al., 2025b), MA-RAG outperforms ODS-v1+Llama3.1-70B (Alzubi et al., 2025b). We believe that the performance gap largely depends on the reasoning model used. Additionally, we highlight that Perplexity Deep Research is a closed-source system without an arXiv paper or public report. These findings suggest that MA-RAG's combination of multi-agent reasoning and web integration is also highly effective for open-domain factual QA.

Table 6: SimpleQA results (Accuracy) for recent systems. * indicates models allowed to access the internet.

| Method | SimpleQA (%) |
|---|---|
| Qwen 2.5 | 9.1 |
| Llama3.1-70B | 20.4 |
| Claude 3.5 Sonnet | 28.9 |
| GPT-4o | 40.1 |
| DeepSeek-R1 | 82.4 |
| Perplexity Deep Research* | 93.9 |
| ODS-v1+Llama3.1-70B* | 83.4 |
| ODS-v2+DeepSeek-R1* | 88.3 |
| ODS-v1+DeepSeek-R1* | 87.7 |
| MA-RAG (GPT-4o-mini, web)* | 86.4 |

*Note:* MA-RAG here is evaluated with live Google Search. All other results are as reported in prior work or reproduced using public checkpoints.

## A.4 IMPLEMENTATION DETAILS

We use 8 NVIDIA A6000 GPUs for LLM inference and employ vLLM (Kwon et al., 2023) to enable efficient generation.

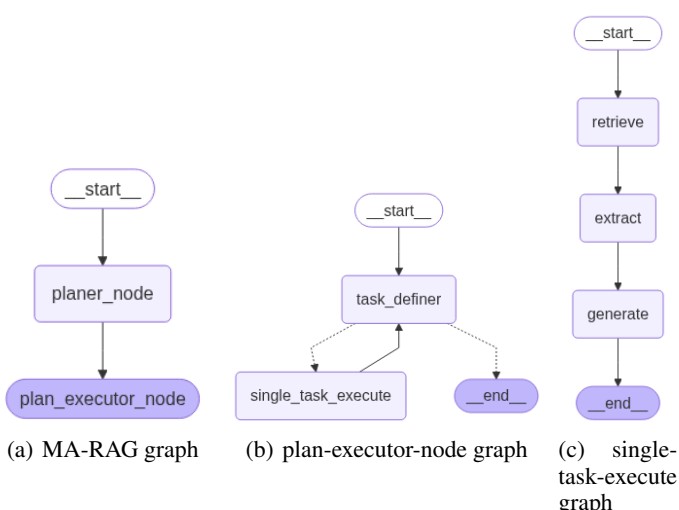

(a) MA-RAG graph    (b) plan-executor-node graph    (c) single-task-execute graph

Figure 4: **MA-RAG** graph representations in Langchain.

## A.5 WORKFLOWS

We implement MA-RAG using LangChain and LangGraph[2], where agents communicate through structured JSON messages. Each agent is represented as a node in the graph, and edges determine the next agent to execute based on the current state. The overall graph structure representing the architecture of MA-RAG is shown in Figure 4. For modularity and clarity, we define separate subgraphs for core components, including the planner, the plan executor, and the RAG pipeline.

### A.5.1 AGENT COMMUNICATION

In LangChain's LangGraph framework, multi-agent workflows are modeled as directed graphs where each node corresponds to an agent, and edges define the flow of control based on task outcomes or states. These nodes operate on a shared mutable object called the Graph State, a dictionary that holds the information exchanged across agents. As agents act on this state, they append or modify fields to propagate reasoning, inputs, and outputs throughout the pipeline.

In MA-RAG, we define multiple GraphState schemas using Python's TypedDict to ensure consistency and clarity in communication between agents. Each sub-state corresponds to a key stage in the pipeline. Below we describe each one individually.

**QAAnswerState**. Stores outputs from the QA agent for each subtask.

```python
class QAAnswerState(TypedDict):
    analysis: str
    answer: str
    success: str
    rating: int
```

**PlanState**. Represents the planner's output plan and rationale.

```python
class PlanState(TypedDict):
    analysis: str
    step: List[str]
```

**StepTaskState**. Encodes detailed instructions for individual subtasks.

```python
class StepTaskState(TypedDict):
    type: str
```

---

[2]https://www.langchain.com/

```
3    task: str
```

**PlanSummaryState**. Summarizes results after plan execution.

```
1  class PlanSummaryState(TypedDict):
2      output: str
3      answer: str
4      score: int
```

**PlanExecState**. Captures the full state of executing a plan, including inputs, intermediate outputs, and notes.

```
1  class PlanExecState(TypedDict):
2      original_question: str
3      plan: List[str]
4      step_question: Annotated[List[StepTaskState], operator.add]
5      step_output: Annotated[List[QAAnswerState], operator.add]
6      step_docs_ids: Annotated[List[List[str]], operator.add]
7      step_notes: Annotated[List[List[str]], operator.add]
8      plan_summary: PlanSummaryState
9      stop: bool = False
```

**RagState**. Manages state during single-step RAG execution.

```
1  class RagState(TypedDict):
2      question: str
3      documents: List[str]
4      doc_ids: List[str]
5      notes: List[str]
6      final_raw_answer: QAAnswerState
```

**GraphState**. The top-level state object that coordinates the MA-RAG pipeline.

```
1  class GraphState(TypedDict):
2      original_question: str
3      plan: List[str]
4      past_exp: Annotated[List[PlanExecState], operator.add]
5      final_answer: str
```

Each agent in MA-RAG reads from and writes to specific fields in these structured states. For example, the Planner agent sets the plan, the Step Definer appends to step_question, the Extractor populates step_notes, and the QA agent writes the step_output. This modular design enables interpretable multi-agent reasoning and seamless communication across the pipeline.

A.6    BOARDER IMPACTS

MA-RAG offers a flexible and interpretable framework for retrieval-augmented generation, which may prove valuable in domains requiring accurate, grounded, and explainable information access. Its modular design allows for fine-grained reasoning steps and clearer attribution of retrieved content, which could support applications in fields like education, healthcare, and research. At the same time, as with any system built on large language models, care must be taken when deploying MA-RAG in high-stakes environments. Even with structured reasoning, the generated outputs may reflect limitations of the underlying LLM or retrieved documents, potentially leading to overconfident or misleading conclusions. As the system enables multi-step reasoning and synthesis, ensuring transparency in intermediate steps and incorporating human oversight remain important considerations for responsible use.

### A.7 PROMPT FORMATS

### A.7.1 PLANNER AGENT

System: You are tasked with assisting users in generating structured plans for answering questions. Your goal is to deconstruct a query into manageable, simpler components. For each question, perform these tasks:
*Analysis: Identify the core components of the question, emphasizing the key elements and context needed for a comprehensive understanding. Determine whether the question is straightforward or requires multiple steps to provide an accurate answer.
*Plan Creation:
- Break down the question into smaller, simpler questions by reasoning that lead to the final answer. Ensure those steps are non overlap.
- Ensure each step is clear and logically sequenced.
- Each step is a question to search, or to aggregate output from previous steps. Do not verify previous step.
Notes:
- Put your output in a list of string, each string describe a sub-task
User: {Question}

### A.7.2 STEP DEFINER AGENT

System: Given a plan, the current step, and the results from finished steps, decide the task for this step.Output the type of task and the query. The query need to be in detail, include all of information from previous step's results in the query if it maked, especially for aggregate task. Be concise.
User:
Plan: {plan}
Current step: {cur_step}
Results of finished steps:
{memory}

### A.7.3 EXTRACTOR AGENT

System: Summarize and extract all relevant information from the provided passages based on the given question. Remove all irrelevant information. Think step-by-step.
**Identify Key Elements**: Read the question carefully to determine what specific information is being requested.
**Analyze Passages**: Review the passages thoroughly to find any segments that contain information relevant to the question.
**Extract Relevant Information**: Highlight or note down sentences, phrases, or words from the passages that relate to the question.
**Remove Irrelevant Details**: Ensure that all extracted information is relevant to the question, eliminating any unnecessary or unrelated content.
Output Format
- Output a list of notes. Each note contains related information from the passage, and each note is clear, standalone.
Notes
- Avoiding any irrelevant details.
- If a piece of information is mentioned in multiple places, include it only once.
- If there are no related information, output: No related information from this document.
User:
Passage: passage
Query: question?

### A.7.4 QUESTION-ANSWERING AGENT

System: You are an assistant for question-answering tasks. Use the following process to deliver concise and precise answers based on the retrieved context.
1. Analyze Carefully: Begin by thoroughly analyzing both the question and the provided context.
2. Identify Core Details: Focus on identifying the essential names, terms, or details that directly answer the question. Disregard any irrelevant information.
3. Provide a Concise Answer:
- Remove redundant words and extraneous details.
- Present the answer by listing only the necessary names, terms, or very brief facts that are crucial for answering the question.
4. Clarity and Accuracy: Ensure that your answer is clear and maintains the original meaning of the information provided.
5. Consensus: If the contexts are not consensus, pick one which is the most logical, consensus, or confident.
User:
Retrieved information: context
Question: question

## A.8 CASE STUDY

### A.8.1 HOTPOTQA

Table 7: A case study on the HotpotQA dataset. The blue texts represent the ground truth answer or correct outputs, while red texts denote distractors or wrong outputs, and green texts stand for evidences. Note that none of the retrieved documents contain the correct answer directly. All experiments are conducted under the same setting, using the same corpus (Karpukhin et al., 2020) and the same retrieval model (Zhang et al., 2024b).

| | |
|---|---|
| **Q**: What role did Thomas Doherty play in sitcom based on the Israeli series "North Star"? ***Correct Answer***: Sean | |
| **MA-RAG 70B** | **Plan**: ["Identify the sitcom based on the Israeli series North Star.", "Determine the character role played by Thomas Doherty in that sitcom."] 
 **Step 1** (Identify the sitcom based on the Israeli series North Star.) 
 **Step query**: What is the name of the sitcom that is based on the Israeli series 'North Star'? 
 **Retrieved DocIDs**: 19587422, 19587424, 14539692, 19587423, 17716638 
 **Extracted Information**: *doc-19587422*: [No related information from this document.], *doc-19587424*: [No related information from this document.], *doc-14539692*: [No related information from this document.], *doc-19587423*: [The sitcom based on the Israeli series 'North Star' is titled "The Lodge." This is evidenced by the passage stating, "The British Disney Channel produces an adapted under the title 'The Lodge' (working title: 'North Star')".], *doc-17716638*: [No related information from this document.] 
 **Step Answer**: The question asks for the name of a sitcom that is based on the Israeli series 'North Star'. The relevant document (doc-19587423) clearly states that the sitcom is titled 'The Lodge'. The other documents do not provide any relevant information. Therefore, the answer is straightforward based on the retrieved context. 
 **Step 2** (Determine the character role played by Thomas Doherty in that sitcom.) 
 **Step query**: What character role does Thomas Doherty play in the sitcom 'The Lodge'? 
 **Retrieved DocIDs**: 20322850, 19608218, 20322852, 20322851, 19608221 
 **Extracted Information**: *doc-20322850*: [Thomas Doherty plays the character role of Sean in the British musical comedy-drama, The Lodge. This is directly stated in the passage where it mentions his best-known roles.], *doc-19608218*: [ Thomas Doherty stars in the role of a character in the TV series The Lodge.The passage does not specify the name of his character or the details of his role], *doc-20322852*: [Thomas Doherty plays the role of Sean in the sitcom The Lodge.This is evidenced by the passage stating, öffered the role of Sean in 'The Lodge'.], *doc-20322851*: [No related information from this document], *doc-19608221*: [No related information from this document] 
 **Step Answer**: The question asks for the character role of Thomas Doherty in 'The Lodge'. The relevant documents consistently indicate that he plays the character named Sean. Therefore, the answer is straightforward based on the retrieved context. 
 **Answer**: Sean |
| **Llama3-70B** | **Answer**: Thomas Doherty played the role of Jasper in the sitcom "High Fidelity", which is based on the Israeli series "North Star." |
| **ChatQA 70B** | **Query**: What role did Thomas Doherty play in sitcom based on the Israeli series "North Star"? 
 **Retrieved Docs**: 
 **Doc-20322850**: Thomas Doherty (actor) Thomas Doherty (born 21 April 1995) is a Scottish actor, best known for his roles as Harry Hook in Disney Channel's "Descendants 2" and as Sean in the British musical comedy-drama, "The Lodge". Doherty was born and raised in Edinburgh, Scotland. He has an older brother and younger sister, and his parents were both in the banking business. Doherty began acting at the age of 5 and also played football. Doherty joined Lothian Youth Arts and Musicals Company and played a variety of roles, including his debut as Disney's Aladdin. Doherty attended Royal High School in Edinburgh. 
 **Doc-8474610**: Tommy Doherty Thomas Edward Doherty (born 17 March 1979 in Bristol) is a Northern Ireland international footballer who plays as a midfielder. Doherty started his career at Bristol City, making 189 league appearances in nine years before moving to Queens Park Rangers in 2005. He joined Wycombe on loan from QPR in September 2006 and was named in the PFA League Two Team of the Year for the 2006–07 season. He was also named the supporters official player of the year. In the summer of 2007 Wycombe tried to sign him but for many reasons a deal was never agreed. 
 **Doc-12942841**: them. The show was canceled after one season owing to poor ratings. She also starred in the popular British sitcom ""Bo! in the USA"", a brain child of Leigh Francis. In the show, she plays herself being randomly harassed by Avid Merrion (Francis), who claims they are lovers. The show aired in October 2006 on the British Channel 4, and she appeared in several episodes. During 2007, Doherty appeared in two television projects. She first appeared in ""Kiss Me Deadly: A Jacob Keane Assignment"" for the Here TV network and followed up with a starring role in the holiday film 
 **Doc-1423761**: to the stage, making a disastrous start in John Phillips' ""Man on the Moon"" (1975). In 1977, he returned to his birthplace, Halifax, Nova Scotia, and started playing legitimate roles, including Shakespeare, at the Neptune Theatre under the tutelage of John Neville. This led to television work, beginning with a variety program, ""Denny's Sho*"", which ran for one season in 1978. He went on to host and voice parts in the children's program, ""Theodore Tugboat"", and to act in various series, including twenty-two episodes of the drama ""Pit Pony"". Doherty also performed with the New Mamas and the Papas (see 
 **Doc-7852137**: Tom Doherty Tom Doherty (born April 23, 1935) is an American publisher, and the founder of science fiction and fantasy book publisher Tor Books. After working as a book salesman in the 1950s and 1960s, Doherty became publisher of Tempo Books in 1972; in 1975, he became, in addition, publisher of another company also owned by Grosset Dunlap, the science fiction imprint Ace Books. In 1980 he left Ace to found his own company, Tor Books. Tor became a subsidiary of St. Martin's Press in 1987; both are now subsidiaries of Holtzbrinck Publishers, ultimately owned by the Macmillan Publishers. 
 **Answer**: Thomas Doherty played the role of Harry Hook in the sitcom based on the Israeli series North Star. |
| **ReAct (70B)** | **Query**: What role did Thomas Doherty play in sitcom based on the Israeli series "North Star"? 
 **AI Messages**: Retrieve(query="Thomas Doherty sitcom North Star role") 
 **Retrieve documents**: 
 **Doc-20322850**: Thomas Doherty (actor) Thomas Doherty (born 21 April 1995) is a Scottish actor, best known for his roles as Harry Hook in Disney Channel's "Descendants 2" and as Sean in the British musical comedy-drama, "The Lodge". Doherty was born and raised in Edinburgh, Scotland. He has an older brother and younger sister, and his parents were both in the banking business. Doherty began acting at the age of 5 and also played football. Doherty joined Lothian Youth Arts and Musicals Company and played a variety of roles, including his debut as Disney's Aladdin. Doherty attended Royal High School in Edinburgh. 
 **Doc-1423761**: to the stage, making a disastrous start in John Phillips' ""Man on the Moon"" (1975). In 1977, he returned to his birthplace, Halifax, Nova Scotia, and started playing legitimate roles, including Shakespeare, at the Neptune Theatre under the tutelage of John Neville. This led to television work, beginning with a variety program, ""Denny's Sho*"", which ran for one season in 1978. He went on to host and voice parts in the children's program, ""Theodore Tugboat"", and to act in various series, including twenty-two episodes of the drama ""Pit Pony"". Doherty also performed with the New Mamas and the Papas (see 
 **Doc-18484135**: After high school, Doherty attended The MGA Academy of Performing Arts in Edinburgh, Scotland. He graduated after studying music theatre at MGA Academy in July 2015 and immediately signed a contract with Olivia Bell Management in London. Doherty was trained in acting, singing, and various types of dance including contemporary, hip hop, jazz, tap and ballet. After graduating from The MGA Academy of Performing Arts in the summer of 2015, Doherty began waiting tables in Edinburgh at Tigerlilly restaurant. On his days off, Doherty split his time between auditioning for ""The Lodge"" and working with the Edinburgh Fringe. Doherty was 
 **Doc-8474610**: Tommy Doherty Thomas Edward Doherty (born 17 March 1979 in Bristol) is a Northern Ireland international footballer who plays as a midfielder. Doherty started his career at Bristol City, making 189 league appearances in nine years before moving to Queens Park Rangers in 2005. He joined Wycombe on loan from QPR in September 2006 and was named in the PFA League Two Team of the Year for the 2006–07 season. He was also named the supporters official player of the year. In the summer of 2007 Wycombe tried to sign him but for many reasons a deal was never agreed. 
 **Doc-2573069**: of Harbour Master, as well as the voice-overs of the characters, in ""Theodore Tugboat"", a CBC Television children's show chronicling the ""lives"" of vessels in a busy harbour loosely based upon Halifax Harbour. In 1999, he also played Charley McGinnis in 22 episodes of the CBC Television series ""Pit Pony"". In 2004, Doherty appeared on Sharon, Lois Bram's 25th Anniversary Concert special, ""25 Years of Skinnamarink"", that aired on CBC on January 1, 2004. He sang two songs with the trio: "California Dreamin" and "Who Put the Bomp?" One of his last appearances was in the Canadian TV series 
 **Answer**: Thomas Doherty played the role of Brandon in the sitcom based on the Israeli series "North Star," which is the series "High Desert". |

### A.8.2 2WIKIMQA

Table 8: A case study on the 2WikimQA dataset. The blue texts represent the ground truth answer or correct outputs, while red texts denote distractors or wrong outputs, and green texts stand for evidences. Note that none of the retrieved documents contain the correct answer directly. All experiments are conducted under the same setting, using the same corpus (Karpukhin et al., 2020) and the same retrieval model (Zhang et al., 2024b).

| | |
|---|---|
| **Q**: When was the company that published Woman'S Century founded? **Correct Answer**: October 27, 1893 | |
| **MA-RAG 70B** | **Plan**: ["Identify the publisher of 'Woman's Century'.", "Determine the founding date of the identified publisher."] 
 **Step 1** (Identify the publisher of 'Woman's Century'.) 
 **Step query**: Identify the publisher of the magazine 'Woman's Century' 
 **Retrieved DocIDs**: 18191569, 18191576, 3964891, 12413254, 9717742 
 **Extracted Information**: *doc-18191569*: [The magazine Woman's Century was published by the National Council of Women of Canada (NCWC). This is evidenced by the statement: Woman's Century was the official organ of the National Council of Women of Canada (NCWC).], *doc-18191576*: [The magazine 'Woman's Century' was published by the National Council of Women of Canada (NCWC). This is evidenced by the statement: Woman's Century was the official organ of the National Council of Women of Canada (NCWC).], *doc-3964891*: [No related information from this document.], *doc-12413254*: [The magazine Woman's Century was published by the NCWC (National Council of Women of Canada). This is evidenced by the phrase Between 1914 and 1921 the NCWC published the monthly journal 'Woman's Century'.], *doc-9717742*: [No related information from this document.] 
 **Step Answer**: National Council of Women of Canada (NCWC) 
 **Step 2** (Determine the founding date of the identified publisher.) 
 **Step query**: What is the founding date of the National Council of Women of Canada (NCWC)? 
 **Retrieved DocIDs**: 12413249, 12413257, 17476996, 6878896, 20577251 
 **Extracted Information**: *doc-12413249*: [The National Council of Women of Canada (NCWC) was founded on October 27, 1893, in Toronto, Ontario. This is relevant as it directly answers the question regarding the founding date of the organization.], *doc-12413257*: [No related information from this document.], *doc-17476996*: [The National Council of Women of Canada (NCWC) was established in 1894, as indicated by the passage stating, "the Local Council was established in 1894 as the local chapter of the National Council of Women of Canada (NCWC)."], *doc-6878896*: [The National Council of Women of Canada was founded in 1893. This information is directly stated in the passage.], *doc-20577251*: [No related information from this document] 
 **Step Answer**: The question asks for the founding date of the National Council of Women of Canada (NCWC). The relevant documents indicate conflicting information: one states it was founded on October 27, 1893, while another mentions 1894. However, multiple documents confirm the 1893 date, making it the more reliable answer. 
 **Answer**: October 27, 1893 |
| **Llama3-70B** | **Answer**: The company that published Woman's Century, initially called the 'Woman's Century Club', later Woman's Century Magazine or just 'Woman's Century', is uncertain but it's known that Woman's Century Club itself predates the magazine as the club was formed in 1891, in Jacksonville, Florida. |
| **ChatQA 70B** | **Query**: When was the company that published Woman's Century founded? 
 **Retrieved Docs**: 
 **Doc-18191576**: Citations Sources Woman's Century Woman's Century was the official organ of the National Council of Women of Canada (NCWC), published in Toronto between 1913 and 1921. The magazine was relatively conservative, and supported World War I, but stood up for women's rights and universal suffrage. The founder of "Woman's Century" was Jessie Campbell MacIver. She had come to Canada from Scotland with her husband, a lawyer, and five children. She became involved in the National Council of Women. The first issue of "Woman's Century" appeared in May 1913. It was largely produced out of MacIver's home, with the help of 
 **Doc-18191569**: Woman's Century Woman's Century was the official organ of the National Council of Women of Canada (NCWC), published in Toronto between 1913 and 1921. The magazine was relatively conservative, and supported World War I, but stood up for women's rights and universal suffrage. The founder of "Woman's Century" was Jessie Campbell MacIver. She had come to Canada from Scotland with her husband, a lawyer, and five children. She became involved in the National Council of Women. The first issue of "Woman's Century" appeared in May 1913. It was largely produced out of MacIver's home, with the help of 
 **Doc-3964891**: The Century Magazine The Century Magazine was first published in the United States in 1881 by The Century Company of New York City, which had been bought in that year by Roswell Smith and renamed by him after the Century Association. It was the successor of "Scribner's Monthly Magazine" and ceased publication in 1930. In 1921, "The Cambridge History of English and American Literature" summarized the early history of the magazine: The magazine was very successful during the 19th century, most notably for the aforementioned series of articles about the American Civil War, which ran for three years during the 
 **Doc-12413254**: also authorized through race-based arguments that linked White women's enfranchisement to the need to protect the nation from "racial degeneration." Between 1914 and 1921 the NCWC published the monthly journal "Woman's Century". The purpose was to educate women about public issues and the reforms that were needed, and to provide a forum for discussion by different women's groups. The title page described it as "A journal of education and progress for Canadian women." The monthly journal was modeled on successful British and American feminist periodicals. It was one of the very few women's rights journals published in Canada. In 1918, 
 **Doc-18227076**: eight-page weekly paper created and managed by the Women's Centennial Executive Committee of Philadelphia. It was "devoted to the Industrial interest of women" by promoting choice, equal pay for equal work, and greater financial and social autonomy. The New Century Club was a woman's organization that was established in 1877 to improve the lives of women. It had committees for working women, municipal affairs and self-education. Rather than portraying the opinions of the "radical" viewpoints of some of its members, the Hallowell, its first president, said that they only "whispered... [the] logic of suffrage." As the organization evolved, the moved 
 **Answer**: The company that published Woman's Century was founded in 1881. |
| **ReAct (70B)** | **Query**: When was the company that published Woman's Century founded? 
 **AI Messages**: Retrieve(query="Woman's Century publication company founding date") 
 **Retrieve documents**: 
 **Doc-18191576**: Citations Sources Woman's Century Woman's Century was the official organ of the National Council of Women of Canada (NCWC), published in Toronto between 1913 and 1921. The magazine was relatively conservative, and supported World War I, but stood up for women's rights and universal suffrage. The founder of "Woman's Century" was Jessie Campbell MacIver. She had come to Canada from Scotland with her husband, a lawyer, and five children. She became involved in the National Council of Women. The first issue of "Woman's Century" appeared in May 1913. It was largely produced out of MacIver's home, with the help of 
 **Doc-18191569**: Woman's Century Woman's Century was the official organ of the National Council of Women of Canada (NCWC), published in Toronto between 1913 and 1921. The magazine was relatively conservative, and supported World War I, but stood up for women's rights and universal suffrage. The founder of "Woman's Century" was Jessie Campbell MacIver. She had come to Canada from Scotland with her husband, a lawyer, and five children. She became involved in the National Council of Women. The first issue of "Woman's Century" appeared in May 1913. It was largely produced out of MacIver's home, with the help of her husband 
 **Doc-3964891**: The Century Magazine The Century Magazine was first published in the United States in 1881 by The Century Company of New York City, which had been bought in that year by Roswell Smith and renamed by him after the Century Association. It was the successor of "Scribner's Monthly Magazine" and ceased publication in 1930. In 1921, "The Cambridge History of English and American Literature" summarized the early history of the magazine: The magazine was very successful during the 19th century, most notably for the aforementioned series of articles about the American Civil War, which ran for three years during the 
 **Doc-18227076**: eight-page weekly paper created and managed by the Women's Centennial Executive Committee of Philadelphia. It was "devoted to the Industrial interest of women" by promoting choice, equal pay for equal work, and greater financial and social autonomy. The New Century Club was a woman's organization that was established in 1877 to improve the lives of women. It had committees for working women, municipal affairs and self-education. Rather than portraying the opinions of the "radical" viewpoints of some of its members, the Hallowell, its first president, said that they only "whispered... [the] logic of suffrage." As the organization evolved, the moved 
 **Doc-12741329**: in 1847 and the Philadelphia Female Anti-Slavery Society in the 1850s. She helped found the Pennsylvania Woman Suffrage Association in 1869, and she was its first corresponding secretary. At the Philadelphia Centennial Exposition of 1876, Turner was a leader of the Women's Congress and distributed the newspaper "New Century for Women" that she wrote and edited at the Women's Pavilion. The New Century Club women's club was founded in Philadelphia in 1877 following a stirring paper that Turner delivered at the Women's Congress. Turner was the president from 1879 to 1881 and the first corresponding secretary of the literary, social 
 **Answer**: The company that published Woman's Century was founded in 1881 in New York City. |

## A.9 LLM Usage Declaration

We declare that large language models (LLMs) were used in limited and specific capacities in this work. Specifically, LLMs such as LLaMA3 (8B and 70B) and GPT-4o-mini were used as components within the MA-RAG framework for retrieval-augmented reasoning. In addition to framework development, LLMs were employed for grammar checking and language refinement of the manuscript. All core technical contributions, experimental design, analysis, and conclusions presented in this work are entirely our own. The use of LLMs did not influence the scientific methodology, results interpretation, or technical contributions of this research.

