# OpenReview forum: "MA-RAG: Multi-Agent Retrieval-Augmented Generation via Collaborative Chain-of-Thought Reasoning"
_ICLR.cc/2026/Conference — Submitted to ICLR 2026_

### Official Review · Reviewer_V6kz · 2025-10-30

**Soundness:** 2
**Presentation:** 2
**Contribution:** 2
**Rating:** 2
**Confidence:** 4

**Summary:**

The paper proposes MA-RAG, a training-free multi-agent retrieval-augmented generation (RAG) framework that integrates four types of agents, including Planner, Step-Definer, Extractor, and QA, along with a retrieval module. The system employs Chain-of-Thought driven collaboration among agents to perform a multi-step process of decomposition, retrieval, evidence extraction, and answer generation. The authors claim that MA-RAG outperforms strong baselines of the same model size on benchmarks such as NQ, HotpotQA, 2WikiMQA, TriviaQA, and FEVER, while also generalizing effectively to the medical domain. They further report the average reasoning steps, component ablation studies, and limited latency statistics.

**Strengths:**

1. The paper treats RAG as a multi-stage systematic reasoning process rather than a typical three-step pipeline of retrieval, augmentation, and generation in traditional RAG. It explicitly exposes intermediate reasoning and evidence filtering through a multi-agent mechanism with on-demand invocation.

2. The evaluation is conducted on multiple datasets with clear and interpretable visualizations, covering both single-hop and multi-hop QA as well as fact verification tasks. The authors report EM/Accuracy as the main metric and provide component-level ablations along with analyses of agent capacity sensitivity.

**Weaknesses:**

1. **Limited innovation and unclear distinction from prior agentic RAG work.**  The proposed framework resembles a modularized version of training-free iterative systems such as Search-o1 [1] or Search-r1 [2], where sub-query decomposition, retrieval, reasoning-in-document, and answering are simply distributed among multiple agents. The contribution appears to be more of an engineering refactor through prompt engineering and role separation rather than a methodological advancement in agentic RAG. The paper does not cite or compare against these widely recognized systems.

[1] Search-o1: Agentic Search-Enhanced Large Reasoning Models

[2] Search-R1: Training LLMs to Reason and Leverage Search Engines with Reinforcement Learning

2. **Unclear necessity of multiple agents.** Each module in the framework could be handled by a single LLM under different instructions, which might even improve global coherence and reduce communication overhead. **The paper does not provide an ablation comparing this single-model unified setup against the multi-agent version to justify why splitting into multiple modules is beneficial.**

3. **Lack of adaptivity in multi-hop retrieval**. In multi-hop QA, **retrieval failure** frequently occurs when an initial sub-query fails to hit relevant evidence, requiring adaptive query reformulation and repeated retrieval. In MA-RAG, the Planner generates a fixed plan at the start, **without incorporating feedback from failed retrieval steps**. The chain of reasoning should evolve dynamically with retrieval outcomes, yet here it remains static and predetermined, which limits robustness in real-world multi-hop scenarios.

4. **Coherence and decomposition accuracy.** It remains unclear how the framework ensures inter-agent coherence across modules, given that the approach is training-free. Without task-specific finetuning, can the Planner reliably decompose complex queries into meaningful sub-queries across different domains?

5. Limited evaluation metrics. The study reports only end-task accuracy (EM/Acc) and omits important diagnostic and process-level metrics such as retrieval hit rate @k, evidence coverage ratio, cross-step consistency with LLM-as-Judge, planner success rate, and Extractor precision/recall. These are crucial to understand which stage of the pipeline contributes most to the performance gain or failure.

**Questions:**

1. Why is a multi-agent design necessary if a single LLM could handle all steps with proper instructions? Could the authors provide an ablation for calling the single agent for conducting all the steps in MA-RAG?

2. Can the Planner dynamically revise its plan after retrieval failures, or is the reasoning chain always fixed?

3. How is coherence maintained between modules without training or shared memory?

4. Could the authors report diagnostic metrics (retrieval hit@k, cross-step consistency with LLM-as-Judge, etc.) to better explain where improvements come from?

---

### Official Review · Reviewer_mKKs · 2025-10-31

**Soundness:** 3
**Presentation:** 2
**Contribution:** 2
**Rating:** 2
**Confidence:** 4

**Summary:**

The paper proposes MA-RAG, a multi-agent retrieval-augmented generation framework designed to improve reasoning and interpretability in complex information-seeking tasks. Instead of treating retrieval, augmentation, and generation as isolated modules, MA-RAG introduces four specialized agents, Planner, Step Definer, Extractor, and QA, to coordinate query decomposition, retrieval, evidence extraction, and answer synthesis through collaborative chain-of-thought reasoning.
Experiments on multiple QA benchmarks show improved results over several baselines, even without fine-tuning, and the authors emphasize modular interpretability and generalization across domains.

**Strengths:**

- The paper is well-structured and clearly written, with detailed illustrations explaining the multi-agent workflow.
- The idea of decomposing the RAG pipeline into distinct reasoning agents is conceptually sound.
- The paper conducts extensive benchmarking across multiple datasets and scales.

**Weaknesses:**

- The motivation of MA-RAG is not clearly articulated beyond being a conceptual adaptation of existing retrieval-augmented generation (RAG) pipelines into a multi-agent form. The introduction mainly highlights general RAG challenges (ambiguity, multi-hop reasoning) but does not provide a concrete insight into why a multi-agent decomposition fundamentally improves these issues beyond modular orchestration. MA-RAG largely repackages these ideas under a multi-agent setting without introducing new algorithmic reasoning mechanisms, objectives, or theoretical insights.
- Comparable multi-agent RAG or task-planning systems already exist such as  MetaGPT, AgentVerse, and AutoAgents, which support modular role specialization and chain-of-thought coordination. MA-RAG does not present measurable advances over these frameworks in terms of reasoning efficiency, dynamic agent scheduling, or learning-based collaboration. The “training-free” aspect is not new either.
- The paper increased latency and token overhead. There is no systematic analysis or quantitative trade-off study. For instance, runtime cost per agent invocation, total token usage. Without detailed cost–performance curves or resource profiling, it is difficult to assess the practicality of MA-RAG compared to simpler baselines

**Questions:**

- How does MA-RAG handle inconsistent or conflicting evidence across multiple retrieved documents?

- Could the authors provide quantitative latency / token-cost comparisons? To what extent do the reported performance gains stem from increased prompt length or token count, rather than genuine reasoning improvements?

- Is there any mechanism for inter-agent feedback or learning beyond static prompting?

---

### Official Review · Reviewer_J7ap · 2025-11-01

**Soundness:** 2
**Presentation:** 2
**Contribution:** 2
**Rating:** 4
**Confidence:** 2

**Summary:**

This paper proposes MA-RAG, a training-free multi-agent framework for Retrieval-Augmented Generation (RAG). Instead of relying on monolithic fine-tuned retrievers or generators, MA-RAG decomposes the RAG pipeline into specialized agents, including Planner, Step Definer, Extractor, and QA Agent, that collaboratively perform query decomposition, retrieval, evidence extraction, and synthesis through chain-of-thought reasoning.

**Strengths:**

Introduce a structured, modular decomposition of RAG with collaborative reasoning and on-demand agent invocation, which is distinct from prior iterative or monolithic designs.

Experiments across multiple benchmarks and domains with robust ablations to validate each design component.

**Weaknesses:**

Multi-agent coordination introduces additional latency and token usage. Although discussed in Section 4.3, quantitative runtime–cost analysis (beyond response time) is limited.

Current evaluation focuses on QA; broader testing (e.g., long-form summarization, reasoning-heavy retrieval) would demonstrate wider applicability.

The relation to other multi-agent LLM coordination systems (e.g., MetaGPT, AgentVerse) could be expanded to highlight MA-RAG’s distinct innovations.

**Questions:**

How sensitive is performance to the number of reasoning steps planned by the Planner agent?

Have the authors explored parallelizing agent execution to mitigate runtime overhead?

Would MA-RAG benefit from lightweight coordination memory (e.g., shared vector store or episodic buffer) between agents?

---

### Official Review · Reviewer_RmFJ · 2025-11-01

**Soundness:** 2
**Presentation:** 3
**Contribution:** 2
**Rating:** 4
**Confidence:** 3

**Summary:**

The paper introduces MA-RAG, a multi-agent framework for retrieval-augmented generation, RAG.
MA-RAG decomposes the RAG process into four cooperating agents, i.e., Planner, Step Definer, Extractor, and QA Agent, and these agents communicate through explicit chain-of-thought (CoT) prompting. The paper presentation is clear, and the experiment is extensive.

**Strengths:**

- The idea of wrapping RAG with a multi-agent system is interesting.
- The paper presentation is clear, especially Section 3.1, along with Table 1 and Table 2, which give the methodological design and empirical evaluation.
- The comparison of Figure 3 is informative.

**Weaknesses:**

- The overall technical novelty and theoretical contribution are not adequate.
- Figures 1 & 2 could be annotated more clearly to show information flow and agent triggers
- The organization should be further polished; the current related work section is long.

**Questions:**

- A central question is whether RAG should be light or heavy, compared to the downstream LLM model. It is interesting to warp RAG as a multi-agent system, then how about using the multi-agent system itself to solve the task? If the RAG should be as large as a multi-agent system, should the word "RAG" be necessary, or is the system already a multi-agent system?
- What is the communication overhead of the MAS, and any analysis?
- Any analysis of how agent errors propagate during the communication?

---

### Meta-Review · Area_Chair_xfE4 · 2025-12-21

**Summary:**

This paper proposes a multi-agent framework for RAG, orchestrating several components such as a planner agent, a step definer agent, a retrieval tool, an extractor agent, and a question answering agent. Experiments on several standard QA datasets (e.g., NQ, TriviaQA) and multi-hop QA benchmarks (e.g., HotpotQA, 2WikiQA) show performance improvement over baselines that do not adopt a multi-agent design.

The following concerns were raised by reviewers and also align with the AC’s independent assessment. The authors did not provide responses.

* Limited novelty or insight: The proposed framework appears to be a trivial extension of prior work, and the distinctions from existing search agent approaches are not clear (V6kz, mKKs, J7ap, RmFJ)
* Lack of justification for multiple agents, as a single agent can plausibly perform the same functions (V6kz, mKKs)
* Lack of adaptivity in multi-hop tasks, as there is no iterative loop for recovering from retrieval failures (V6kz)
* No discussion on latency and token overhead (mKKs, J7ap)
* Narrow evaluation scope (J7ap)
* The paper’s organization and the clarity of several figures could be improved (RmFJ)

**Reviewer Concerns:**

Noted above

**Reviewer Scores:**

Authors did not provide responses

---

### Decision · Program_Chairs · 2026-01-26

Reject